# Analysis of the Sustainability of Fattening Systems for Iberian Traditional Pig Production through a Technical and Environmental Approach

**DOI:** 10.3390/ani11020411

**Published:** 2021-02-05

**Authors:** Javier García-Gudiño, Isabel Blanco-Penedo, Maria Font-i-Furnols, Elena Angón, José Manuel Perea

**Affiliations:** 1Animal Welfare Program, IRTA, 17121 Monells, Spain; isabel.blanco.penedo@slu.se; 2Department of Clinical Sciences, SLU, SE-750 07 Uppsala, Sweden; 3Food Quality and Technology Program, IRTA, 17121 Monells, Spain; maria.font@irta.es; 4Animal Production, UCO, 14071 Córdoba, Spain; eangon@uco.es (E.A.); pa2pemuj@uco.es (J.M.P.)

**Keywords:** local breed, feeding systems, multivariate analysis

## Abstract

**Simple Summary:**

Iberian traditional pig production has been linked to the use of the natural resources of the *dehesa* ecosystem. In the last decades, the Spanish livestock sector has experienced a significant transformation towards the intensification of livestock systems. The intensification of the system combined with the increased demand for high-quality Iberian products resulted in a greater demand for feedstuffs as inputs into the Iberian pig production system. For these reasons, the Iberian pig exploitation in the *dehesa* ecosystem should be studied considering economic and environmental criteria to identify strategies for more sustainable livestock production. From the analyses carried out, the relationship between livestock management and environmental values obtained has been determined. Iberian traditional pig production has room for improvement in terms of economic and environmental values. In order to achieve this, appropriate fattening strategies should be implemented to optimize the use of available resources and improve economic-environmental performance for sustainable development. The importance of exploring sustainable management on this animal system derives because a sustainable Iberian traditional pig production has an important role in maintaining the population in rural areas through livestock activity as an economic engine.

**Abstract:**

At present, two types of fattening are carried out in Iberian traditional pig production. The *montanera* is the fattening system where fatteners are fed on acorns and pasture in the *dehesa*, and *cebo de campo* is the fattening where the pigs are fed on compound feed and natural resources, mainly pasture. The aim of this paper is to analyze Iberian fattening production from an economic and environmental approach in order to identify fattening strategies to increase the sustainability of this traditional livestock activity. Based on technical-economic and environmental variables, the differences between Iberian farms according to the types of fattening were determined using discriminant analysis techniques. The model based on environmental variables showed a greater predictive ability than that found in the model based on technical-economic variables. Consequently, environmental variables can be used as reference points to classify the Iberian farms according to the type of fattening. Furthermore, canonical correlation analysis allowed to study the relationships between both sets of variables, showing that environmental values had a strong correlation with technical-economic variables. The results of this study show that it is possible to improve the sustainability of Iberian traditional pig production through fattening strategies in both types of fattening.

## 1. Introduction

Iberian traditional pig production has a significant role in the Spanish pig industry, where intensive systems are predominant [1]. The sustainability of traditional livestock systems being less competitive than conventional systems [2] has been possible by added values offered [3,4,5]. This greater product differentiation and value-adding over time has shifted the interest to outdoor pig production systems by consumers [6] with a more critical view towards intensive livestock production [7]. Therefore, according to this reasoning, different production managements can be found in the actual Iberian pig sector [8], increasing the number of fatteners produced annually [9].

At present, two different fattening types are developed in the *dehesa* and count with a specific legislation [8]: *montanera* and *cebo de campo*. The *montanera* is the traditional fattening type based on local natural resources under extensive management [10]. The limitation of *dehesa* hectares [11] form the basis of the human intervention that increases the use of external inputs in Iberian pig production. For this reason, the *cebo de campo* emerged as the fattening characterized by the use of natural resources combined with compound feed. Decreased use of natural resources in *cebo de campo* has resulted in an exponential increase in the number of fatteners produced per year to over one million in the Iberian traditional pig production area [9]. However, the number of animals produced between the two fattening types is approximately the same.

The perfect adaptation of Iberian breed to the *dehesa* ecosystem has promoted the persistence of this local breed and its productive system [12]. However, the traditional image of the Iberian pig in the *dehesa* has been denatured because of the large increase of animals produced [13]. This higher demand has led to environmental stress at the *dehesa*, which endangers the traditional livestock system. The overexploitation of the agroecosystem can lead to a series of negative consequences such as soil erosion [14] or decrease in oak regeneration [15].

Because Iberian pig and the *dehesa* ecosystem constitute a real symbiosis [6], pig production and the agroforestry system must be assessed in a combined approach. In this way, pig production could be economically viable and environmentally friendly [16], both crucial for the preservation of the *dehesa* and the future of livestock as economic engine [17] to sustain the population in rural areas [6]. Previous works on environmental assessment showed a completely different conclusion of that resulted from an economic analysis in livestock systems [18]. Studies on livestock production in the *dehesa* have addressed the technical-economic [2,19,20] and environmental assessment [5,21,22] on singles pieces in general. Therefore, it is of great importance to evaluate the actual situation of Iberian pig production in the *dehesa* through different economic and environmental analytical approaches aiming to balance the economic and environmental pillars of the sustainability of the traditional Iberian pig production.

The purposes of this paper can be described as (i) exploring the differences in Iberian farms according to the fattening types based on technical-economic and environmental approach, (ii) identifying the characteristics that can be reference points to differentiate the fattening types, and (iii) proposing strategies to improve sustainability of Iberian traditional pig production.

## 2. Materials and Methods

### 2.1. Description of the Pig Fattening at the Iberian Traditional Pig Production

The growing period in this traditional system is based on extensive or semi-extensive management from 23 kg to 95–105 kg of live weight [23]. The growers are fed with compound feed and they consume different natural resources depending on season [24].

Differences of management are mainly found in the finishing or fattening period (Table 1). According to the Spanish legislation [8], the finishing period can be appointed as *montanera* or *cebo de campo* in the *dehesa*. The types of fattening are defined by stocking density, feeding, and age at slaughter. According to stocking density, the *montanera* should rear between 0.25 and 1.25 fatteners per hectare depending on the available wooded area. On the other hand, the stocking density is fixed in 15 fatteners per hectare in the *cebo de campo* during the finishing period. Regarding the feeding, fatteners should consume only natural resources (acorn and grass) in the *montanera* while fatteners are fed with compound feed and natural resources available (mainly pastures) in the *cebo de campo*. In terms of the minimum age at slaughter, it is fixed on 14 months in the *montanera*, and 12 months in the *cebo de campo*. In both cases, the fattening period out of the life cycle of the animal must last a minimum of 60 days. With the feeding availability in mind, the *cebo de campo* can be developed throughout the year, while the *montanera* can only occur between October and March due to the availability of acorns in the *dehesa*.

### 2.2. Data Acquisition

Data were collected through questionnaires from 36 farms in the Iberian traditional pig area (SW Spain). Data achieved for this study were farm area, number of animals (pigs and other species), productive (e.g., daily ration, live weight, age at slaughter) and reproductive (e.g., fertility, prolificacy) data, economic and management aspects, inventory (machinery and facilities), and information about other activities (agriculture and livestock).

Environmental variables employed were derived from García-Gudiño et al. [5]. Global warming (GW, kg CO_2_ eq) and land occupation (LO, m^2^·year) were used for the environmental assessment in the present study. Analyses of Life Cycle Assessment (LCA) were performed with Simapro software (version 8.5.2.0, PRé Consultants, Amersfoort, The Netherlands). The functional unit was one kilogram of live weight at farm gate.

Technical and environmental variables are described in Table 2. Some of them are economic variables that are related either to technical or environmental aspects.

### 2.3. Statistical Analysis

Preliminary testing of data was carried out to determine outliers to be discarded before analysis, using the Grubb’s test, and to determine Pearson correlations to avoid variables that presented a correlation coefficient with an absolute value >0.95 [25]. Because data had different measurement units, variables were standardized to zero mean and a unit standard deviation.

Farms were classified into two groups according to the type of fattening: *Montanera* farms (**MF**) if more than 90% of the fattened pigs were certified as Iberian acorn-fed [8], and diversified farms (**DF**) otherwise. In **DF**, Iberian acorn-fed pigs are less than 90% of the pigs fattened on the farm. The rest of the fattened pigs are fattened through the *cebo de campo*. Therefore, the types of fatteners produced (*montanera* or *cebo de campo*) in **DF** are more diverse than in **MF**.

Multivariate analysis techniques were used to analyze differences and similarities in technical and environmental variables among fattening types, and to evaluate the specific relationships between technical and environmental variables. To discriminate between the two groups (**MF**/**DF**), three complementary and sequenced techniques were applied in the following order: canonical discriminant analysis, stepwise discriminant analysis, and discriminant analysis. These techniques have been applied in previous studies on livestock systems [26,27,28,29,30].

Canonical discriminant analysis is a dimension-reduction technique related to principal component analysis and canonical correlation, which gives information about the similarities of the fattening types implemented in Iberian pig farms. It was applied to all the variables described in Table 2. Given a classification character several variables, canonical discriminant analysis derives a set of new variables, called canonical functions, which are linear combinations of the original variables that summarize between-group variation in the data, highlighting their differences [28].

The minimum number of variables able to discriminate between the two groups was obtained after performing a stepwise discriminant analysis on two sets of variables: those related to technical variables, those related to the environmental variables of the fattening types, and those related to both sets of variables. This procedure selects the variables to include in the model based on how much they contribute to decrease Wilks’ λ. In the first step, the most discriminating variable enters into the model, and in subsequent steps, the entry or removal of variables is evaluated according to an entry and remove threshold that was set at 0.05 and 0.10, respectively. To avoid information redundancy, a tolerance level of 0.01 was set. The steps are repeated until no more variables can be entered or removed, or until the maximum number of steps is reached, which was set as twice the number of original variables in each model. The efficiency of the discriminant power of a given model was determined using the Wilks’ λ test of significance. The effective separation of groups was assessed using Mahalanobis distance and the corresponding Hotelling’s T^2^ test [31].

The canonical discriminant analyses were repeated with the selected variables derived from stepwise discriminant analyses to obtain the most plausible canonical functions, and from these, discriminate between the fattening types. The predictive ability of each model was tested using the absolute assignment of individuals to the preassigned group [32].

The second step was to study the existing relationships between technical and environmental variables of the fattening types. Canonical correlation analysis was deemed appropriate because it provides not only the magnitude of the relationships that may exist between groups of variables but also a quantification of the relative contribution of each variable to those relationships [33]. Canonical correlation analysis complements discriminant analysis, because the latter explores only associations between data without explaining why they exist [28].

Canonical correlation analysis is a multivariate analysis method based on the linear relationship between two multidimensional variables, *X* (technical) and *Y* (environmental). The aim of this analysis is to find linear combinations (*U = a^T^X* and *V = a^T^Y*) so that the correlation between *U* and *V* is maximized. Such linear combinations reflect the relationship between both sets of variables [26,34]. The basic principle of canonical correlation analysis is the construction of subsequent pairs of canonical variables (*Ui*, *Vi*), that are linear combinations of the originals, so that each pair is orthogonal to the previous and represents the best explanation of the *Y* set (formed by q dependent variables) with respect to the *X* set (formed by p independent variables) that has not been obtained by the previous pairs [27,35]. All statistical analyses were performed using the XLSTAT© software (procedures: Grubbs test for outliers, Similarity/Dissimilarity matrices, Discriminant analysis, Canonical correlation analysis).

## 3. Results and Discussion

### 3.1. Differentiation of Iberian Fattening Production

Results of the canonical discriminant analysis based on technical and environmental variables are presented in Table 3. The most discriminating variables between the fattening types implemented in Iberian pig farms are noted in Table 3.

From technical variables studied, the most discriminant variables were “LU value”, “*Dehesa* land use”, and “Sows”. Regarding the second component of the analysis, related to environmental variables, the most discriminant variables were “GW value” and “GW”. Considering both sets of variables together, those variables with a greater discriminant ability were “LU value”, “GW”, and “GW value”.

Differences between fattening management types were observed in technical variables (Table 3). The most important and significant difference between Iberian farms is found in “*Dehesa* land use”, influencing in the fattening management. “*Dehesa* land use” increase the natural resources availability when this technical variable moves to higher values. For this reason, **MF** showed higher “*Dehesa* land use” compared to **DF** and it is explained by the higher percentage of fatteners’ acorn and grass-fed during the finishing period. In other technical variables, no significant differences were found due to the great variability shown on **DF** data. Nevertheless, the different types of Iberian farms can be characterized through the results obtained. While **MF** produces higher *montanera* meat production per *dehesa* hectare, **DF** obtains a greater pig meat production per hectare. A higher meat production in **DF** is achieved to a higher pig stocking rate that characterizes the intensification of this type of farm system. On **DF**, the production of pigs through the two coexistent types of fattening (*montanera* and *cebo de campo*) and a higher number of sows both finally increase the pig stocking rate. In contrast, the legal requirement of several hectares of *dehesa* for animal feeding purposes reduces the pig stocking rate in **MF**. The combined condition of a greater number of animals as an output together with a lower “*Dehesa* land use” lead **DF** towards dependence on compound feed because of lower natural resource availability per animal produced. Because of this feed dependency, feedstuffs inputs per hectare are 3.5 times superior on **DF** than on **MF** in this study.

Environmental differences were observed between Iberian farms (Table 3) majorly caused by the management described previously. Intensification of livestock production increases the inclusion of concentrated feed in the diet and decreases the grazing period, causing negative environmental impacts [36]. From LCA, “GW” is lower on **MF** than **DF** which indicates that a greater use of natural resources in **MF** is the best measure for reducing the environmental impacts on livestock activities [5,37,38], since a high number of animals per unit limit the availability of natural resources increasing the consumption of compound feed on **DF**. In contrast, LCA shows a trend towards greater “LO” in **MF** than **DF**. The trend might correspond to the attachment of natural resources on *montanera* that requires a higher area requirement for feeding animals versus a lower land requirement in *cebo de campo* (feedstuffs inputs).

Economic differences were observed between the participant Iberian farms (Table 3). The relationship between the economic value generated and technical variables indicates that **MF** obtains higher income per livestock unit (LU value). The higher income per livestock unit in **MF** is due to a higher price of fatteners *montanera* in the market compared to other fatteners pigs in other livestock systems around the globe [39,40]. In addition, the economic value obtained for 1 kg CO_2_ emitted (GW value) in **MF** is higher than in **DF** because **MF** is based on natural local resources use with the ultimate result of a reduction in GHG emissions [5].

### 3.2. Reference Points in Iberian Fattening Production

The canonical discriminant models obtained from the stepwise discriminant analysis based on technical and environmental variables are presented in Table 4. In both sets of variables, the extracted canonical functions significantly discriminated between the two types of fattening farms (**MF** vs. **DF**; *p* < 0.001, Hotelling’s T2 test). The F-statistics revealed a higher discriminating ability for variables related to environmental performance. Figure 1 also allows seeing the higher variability in **DF** than **MF** which seems reasonable due to the different types of animals produced. This outcome is supported by the Mahalanobis distances among farm groups (Figure 2). The Mahalanobis distances among **MF** and **DF** were 2.10 for technical variables, 2.16 for environmental variables, and 2.46 for both sets. Therefore, the two fattening types studied are distanced because all pairwise distances were significant [30].

Discriminant analysis classified the fattening farms on a preassigned group according to the selected technical or environmental variables (Table 5). The model based on technical and structural variables classified 83.3% of the farms correctly, and the model based on environmental variables correctly classified 97.2% of the participant farms. In addition, 85.7% of classification errors occurred on technical variables, while there was only one misclassification regarding environmental performance. These results indicated that the set of environmental variables discriminate much better than the set of technical variables the differences in management among the two different fattening types. The model based on technical and environmental variables showed a predictive ability equal to that of the model based only on environmental variables. Therefore, the set of environmental variables can be used as reference points to classify the types of fattening carried out on Iberian farms.

Results obtained from canonical correlation analysis are presented in Table 6. The model extracted 58.52% of the variance from the set of structural and technical variables, and 100% of the variance for the set of environmental variables. Canonical correlations for the first and second pair of canonical variables were 0.973 and 0.844, respectively. These values were significant and represented 69.24% of the variability observed in the model.

The correlation structure (Figure 3) showed that environmental performance was strongly correlated with land use, degree of intensification, and feeding practices. The first pair (F1) of canonical variables linked environmental values with land use and degree of intensification (Figure 3), showing that a more intensified fattening system generates a higher economic yield per hectare. The main cause of higher profitability is the increase in the number of animals produced. The production of fatteners *cebo de campo* is carried out in a lower area (15 fatteners pigs per hectare) as stated in the legislation [8], increasing the number of animals by area on more intensive management [37]. To improve profitability, Iberian farms can produce various production cycles of fatteners on *cebo de campo* per year. In contrast, Iberian farms with the exclusive production of fatteners in *montanera* only could fit one productive cycle per year as the ability of the Iberian pig breed to feed on acorns is possible from October to March [41]. Consequently, a greater number of productive cycles in *cebo de campo* increases the economic value per hectare.

The second pair (F2) of the canonical variables linked environmental variables with feeding practices (Figure 3). The best practice to reduce emissions is to increase the proportion of natural resources in animal feeding [5] and to reduce the use of imported feedstuff. For reaching this goal, the fattening systems must optimize the use of the resources of the *dehesa*. As a result, the ratio of fatteners *montanera* in relation to the number of animals produced would increase the feeding through natural resources base. The predominance of fattening *montanera* together with a better price of fatteners *montanera* in the market increases “LU value” in Iberian farms where fatteners *montanera* are produced. Consequently, Iberian farms with a lower production of emissions generate more economic value per environmental unit emitted. According to our results and interpretations, the first combination of standardized canonical variables could be considered a predictable measure of LO, and the second combination could be considered a predictable measure of GW.

The fattening management per se determines the economic and environmental characteristics of the farm unit. **MF** is more environmentally friendly due to extensive fattening management focused on better use of the *dehesa*’s natural resources [6]. **DF** is more profitable due to a more intensive management in the fattening period, increasing the stocking rate and feedstuffs inputs. This interpretation is in line with other studies on Iberian pig production [6] which contributes to show sample representativeness of the participant farms in this study.

### 3.3. Improvements for More Sustainable Iberian Fattening Production

Through the results obtained in the present study, it is possible to elaborate strategies focused on the improvement of the sustainability of the Iberian pig sector in the *dehesa*. Based on the optimal economic and environmental results obtained by the **MF**, the Iberian pig traditional livestock should be oriented towards the production of finishing pigs in *montanera* as a first option. The reason is mainly based on the environmental values obtained for the close attachment to natural resources during fattening *montanera* [5].

The reduction of inputs required by making more efficient use of internal resources can improve the environmental sustainability of livestock activity [42]. For this reason, the **MF** can be more environmentally sustainable through the optimization of the resource-use of the *dehesa* ecosystem. The **MF** should maximize “kg *montanera*” through increased fatteners *montanera* stocking rate in the *dehesa* during the finishing period, still under the framed legislation. To achieve this goal, the reforestations are necessary to increase the number of fatteners *montanera* that are produced. According to Spanish legislation [8], the farm unit could increase from 0.25 to a maximum of 1.25 fatteners *montanera* per hectare depending on the woodland density. As a result of this improvement, the “kg *montanera*” would increase while “LO” per kg of live weight at farm gate would decrease [43]. This way, increased efficiency generates both an improvement in livestock and environmental performances [36].

Although the finishing period in *montanera* should be the first option for fattening pigs, the *cebo de campo* fattening is necessary for several reasons in Iberian traditional pig production at present. For instance, the *cebo de campo* fattening is a valid alternative for the overproduction of piglets that exceeds the capacity of the *dehesa* to fatten pigs with natural resources only [11]. In this way, the surplus of piglets is not converted into an undesirable output. If the *cebo de campo* was more linked to the land, the feed inputs required would have been reduced [42]. This is a more favorable scenario since feed production is the main hotspot for several environmental impacts [36,38,44]. For this reason, adapting feeding strategies and animal management can reduce, to some extent, the environmental impacts [45] of the Iberian traditional pig production. The good management practices can be carried out during the phases of growing and fattening because the Iberian pigs are fed with compound feed and natural resources in both phases.

The results showed that **DF** consumes 3.5 times more feedstuffs inputs than **MF**. A decrease in compound feed consumption reduces the environmental impacts resulted from feed production [16]. For that purpose, optimal use of pasture is an appropriate feeding strategy for extensive systems since outdoor pigs obtain a considerable portion of nutritional requirements from grazing, reducing the daily ration [38]. Furthermore, the integration of pig production into cereal crops is possible [46]. Iberian pigs can graze the cereal crop before the earing phase [47], and the harvested grain can be used as additional feed for Iberian pigs, reducing the number of feed inputs. Another feeding strategy to improve the sustainability of pig production is the use of local feed products [48]. For instance, some authors [38,49,50] investigated the use of local subproducts in swine feed, quantifying a reduction of environmental impacts. In addition, the use of local protein sources in feed production such as sainfoin [51], grain legumes [44], or rapeseed [52], among other alternative sources [53] showed a reduction of the environmental impact of different pig systems and geographical contexts.

## 4. Conclusions

In the conditions of the present work, it is possible to conclude that the Iberian pig production located in the *dehesa* ecosystem shows a great differentiation in technical and environmental aspects according to the type of fattening. The results show that the relationship between technical and environmental variables is strong. Due to this relationship, the classification of Iberian farms according to the type of fattening is possible through environmental variables in a more precise manner.

In the Iberian pig production located in the *dehesa* at present, the two concurrent types of fattening are necessary and complementary. While the fattening *montanera* optimises the use of the natural resources offered by the *dehesa*, being a more eco-friendly livestock production, the fattening *cebo de campo* permits the fattening phase to be carried out when acorns are not seasonally available, resulting in a more profitable pig production. The combined use of fattening *montanera* and *cebo de campo* is the optimal fattening strategy to improve the sustainability in Iberian traditional pig production.

In order to improve the sustainability in Iberian traditional pig production, environmental impacts of these systems may need to be mitigated by good management practices. Further investigations are needed to explore strategies that focus on reducing environmental impacts and increasing profitability at the Iberian farms.

## Figures and Tables

**Figure 1 animals-11-00411-f001:**
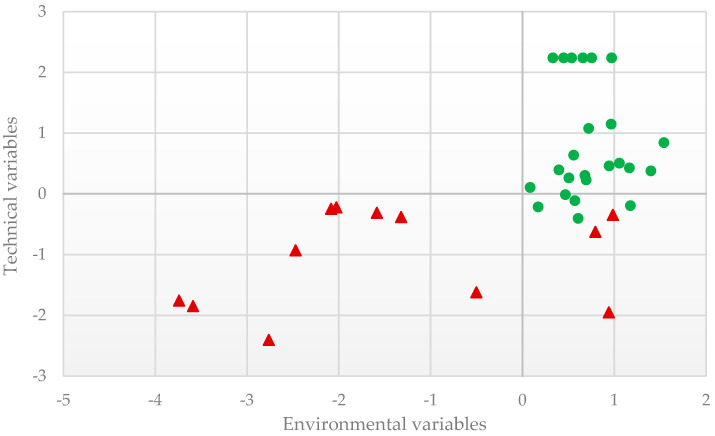
Graphic representation of the results from canonical discriminant analysis for technical and environmental variables, defined by the axes of the first canonical variables (red triangle-Diversified farms; green spot-*Montanera* farms).

**Figure 2 animals-11-00411-f002:**
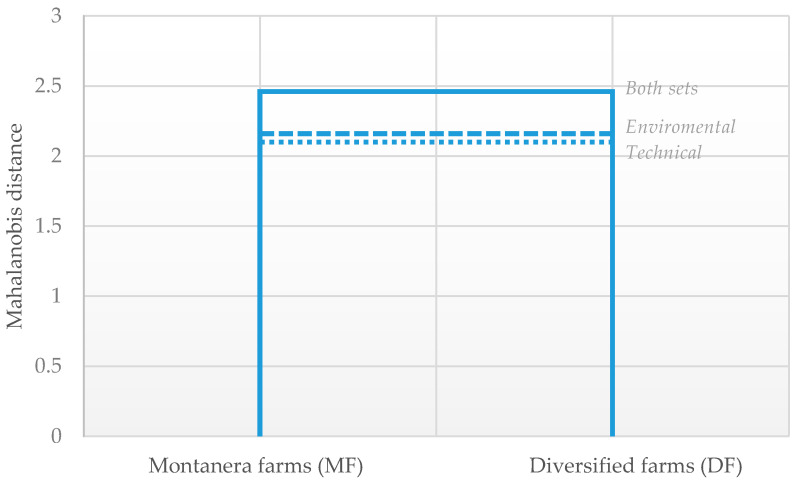
Dendrogram showing technical and environmental variables relationship between Iberian farms.

**Figure 3 animals-11-00411-f003:**
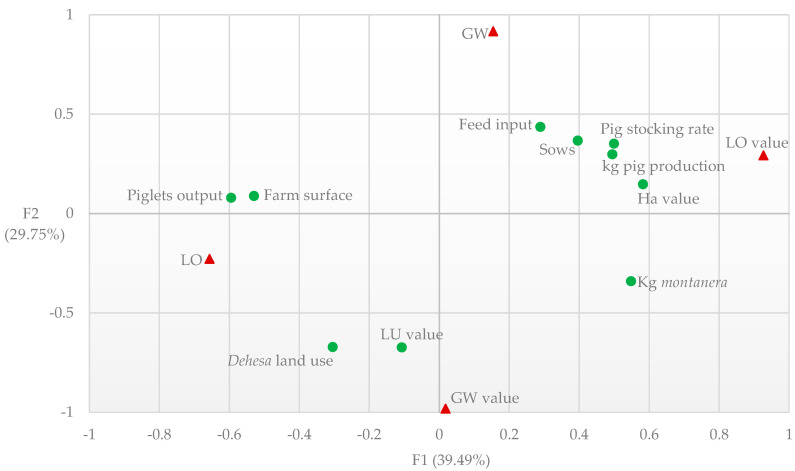
Canonical correlation analysis similarity map determined by the first and second canonical variables for technical (green spot) and environmental variables (red triangle).

**Table 1 animals-11-00411-t001:** Requirements of management in Iberian fattening period according to the legislation (RD 4/2014).

Requirements Fattening Period	*Montanera*	*Cebo de Campo*
Feeding	Natural resources (acorn and grass)	Compound feed and natural resources (grass)
Stocking density	0.25–1.25 animals/ha	15 animals/ha
Minimum duration	60 days	60 days
Minimum age at slaughter	14 months	12 months

**Table 2 animals-11-00411-t002:** Technical and environmental variables used to evaluate fattening types in Iberian farms (*n* = 36).

Variable	Description	Mean	SE ^1^
*Technical variables*			
Farm surface	Total surface area, ha	631.60	104.02
*Dehesa* land use	Utilized *dehesa* area/Total *dehesa* area, %	79.25	5.34
Pig stocking rate	Pig livestock unit per ha, LU/ha	0.12	0.02
Sows	Sows per 100 kg of pig	0.16	0.09
Piglets output	Piglets produced per fattened pig	1.48	0.19
kg *montanera*	kg of live weight from fatteners *montanera* per *dehesa* area, kg/ha	95.70	17.21
kg total pig production	kg of live weight from pig production per total surface area, kg/ha	106.49	17.67
ha value *	Production value per ha, €/ha	298.15	43.93
LU value *	Production value per pig livestock unit, €/LU	2554.4	100.69
Feedstuffs inputs	Animal feedstuffs per ha, kg/ha	212.69	92.77
*Environmental variables*			
GW	Global warming, kg CO_2_ eq	3.75	0.75
LO	Land occupation CML non baseline, m^2^·year	38.72	3.60
GW value *	Production value per GW, €/CO_2_ eq	0.795	0.027
LO value *	Production value per LO, €/m^2^·year	0.092	0.007

* Economic variables relating to technical and environmental aspects. ^1^ SE: standard error. LU: livestock unit. GW: global warming. LO: land occupation.

**Table 3 animals-11-00411-t003:** Results of canonical discriminant analysis with technical and environmental variables.

Variable	*Montanera* Farms (MF)	SE ^1^	Diversified Farms (DF)	SE ^1^	Wilks’ λ	F-Value	*p*-Value	CAN ^2^
*Technical variables*								
Farm surface	658.6	132.4	577.7	171.9	0.996	0.13	0.720	0.088
*Dehesa* land use *	87.17	4.89	63.42	11.75	0.875	4.87	0.034	0.606
Pig stocking rate	0.09	0.02	0.16	0.05	0.950	1.79	0.190	−0.419
Sows *	0.06	0.01	0.37	0.29	0.936	2.33	0.014	−0.349
Piglets output	1.30	0.21	1.86	0.37	0.945	1.98	0.168	−0.305
kg *montanera*	113.5	24.36	60.02	13.13	0.939	2.22	0.145	0.339
kg total pig production	97.06	16.07	125.4	43.02	0.984	0.56	0.458	−0.371
ha value	291.2	48.71	312.1	92.05	0.999	0.49	0.826	−0.346
LU value *	2825.5	72.49	2012.0	186.3	0.586	24.06	<0.001	0.962
Feedstuffs inputs	115.9	43.52	406.3	262.9	0.938	2.25	0.142	−0.152
*Environmental variables*								
GW *	3.41	0.05	4.44	0.27	0.570	25.66	<0.001	−0.953
LO	43.00	4.70	30.15	4.59	0.919	3.00	0.09	0.492
GW value *	0.88	0.01	0.63	0.06	0.486	36.00	<0.001	0.989
LO value	0.09	0.01	0.11	0.01	0.951	1.76	0.193	−0.376

^1^ SE: standard error. ^2^ CAN: correlation of each variable with the canonical function. * Most discriminating variables between fattening types. LU: livestock unit. GW: global warming. LO: land occupation.

**Table 4 animals-11-00411-t004:** Discriminant canonical models for technical and environmental variables.

Model	Variables in the Model	Number of Groups	Wilks’ λ	F-Value	*p*-Value
Technical	Sows, Value LU	2	0.474	18.30	<0.001
Environmental	LO, Value GW	2	0.425	45.36	<0.001
Both sets	Value LU, Value GW	2	0.411	35.99	<0.001

**Table 5 animals-11-00411-t005:** Assignation percentages in the predefined groups and classification errors.

Group	Montanera Farms (MF)	Diversified Farms (DF)
*Technical model*		
Montanera farms	95.83	4.16
Diversified farms	41.66	58.33
Level of error	0.18	0.13
Prior probability	0.50	0.50
*Environmental model*		
Montanera farm	100.00	0.00
Diversified farm	8.33	91.66
Level of error	0.04	0.00
Prior probability	0.50	0.50
*Both sets of variables*		
Montanera farm	100.00	0.00
Diversified farm	8.33	91.66
Level of error	0.04	0.00
Prior probability	0.50	0.50

**Table 6 animals-11-00411-t006:** Canonical correlation analysis on technical and environmental variables.

Factor	Eigen Value	Canonical Correlation	Variability, %	Wilks’ λ	*p*-Value
F1	0.946	0.973	39.49	0.006	<0.001
F2	0.712	0.844	29.75	0.104	<0.001
F3	0.559	0.748	23.36	0.362	0.032
F4	0.178	0.421	7.40	0.822	0.615

## Data Availability

The data presented in this study are available on request from the corresponding author. The data are not publicly available due to project IP rules.

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
