# Peer review of "Analysis of the Sustainability of Fattening Systems for Iberian Traditional Pig Production through a Technical and Environmental Approach"

_animals, 2021, doi:10.3390/ani11020411_

Round 1
Reviewer 1 Report
General comments
The paper deals with the study of differences in Iberian farms according to the fattening types based on technical-economic and environmental approach, as well as with the identification of the characteristics that can be reference points to differentiate the fattening types, and proposing strategies to improve sustainability of Iberian traditional pig production. The aim of the paper is interesting since the integration of technical-economic and environmental issues is necessary in the study of sustainability of livestock systems.
Taking into account the analyses performed and results of the paper I suggest to change the title of the paper. Because although some fattening strategies to improve sustainability of the systems were exposed by the authors at the end of the paper, in my opinion it was not the main objective of the paper.
The paper overall is sound and well written. The introduction section properly defines the problem and the proposed hypothesis taking into account related bibliography.
Methods have to be explained more in detail, it was not clear what was the sequence of the different analyses performed. Authors should clarify this section. Discussion in general is well written and succeed in discussing scientifically the observed findings in a biologically integrated fashion, both within the study as well as relative to results of other scientists.
Specific comments
Abstract
L35: “to study the relationships” instead of “to study of the relationships”
Material and Methods
L112: I suggest to move this sentence “Data were analysed to determine outliers to be discarded before analysis” to “statistical analyses section” and explain what statistic method was used to identify outliers.
L114: please describe all abbreviations the first time they appear in the text. Here and elsewhere
Table 2: define abbreviations in the table footnote. In table 2 mean and SE was shown for the 36 studied farms. It would be of interest to show the mean and SE for each fattening type (MF and DF) and the number of farms in each group.
L139: please indicate the statistical programs or packages used for the discriminant analyses.
L139-140: “These techniques have been applied in previous studies with a focus on dairy sheep systems [25–27], meat goat systems [28] and fishery systems [29)” : this sentence is not adequate for the mat methods section .
L141-146: I think this paragraph is not adequate for the mat methods section, maybe should be include to some extent in the discussion of the results.
L154: it is not clear the order of the performed analyses. In Line 138 the following order was assessed:” canonical discriminant analysis, stepwise discriminant analysis, and discriminant analysis.”, but here seems that first was performed the stepwise discriminant analysis. Please indicate the correct order for the three analyses.
L157: how was it tested?
L165-174: In my opinion it is not necessary to explain what is a CCA analysis in the material and methods section.
L174: Authors indicate the software but also should indicate the procedures.
There are much more technical variables than environmental ones, and the environmental variables are highly correlated between them (i.e. GW and GW value or LO and LO value). These differences could affect the capacity of the models to discriminate between fattening types?. Could exist and overestimation of the discriminant capacity of the environmental model?.
Results and discussion
177-180: please clarify if table 3 showed results of the canonical discriminant analysis or results of stepwise discriminant analysis.
L184: Why is “sows” declared as a discriminant variable if the P-value in table 3 is not significant?
Table 3 : define abbreviations in the table footnote. Every table and figure are independent and abbreviations should be described
L196-206: here authors described numerical changes, because these variables had not a significant P-value
L226-229: please do not discuss non-significant values
L231: does it refers to the results of the canonical discriminant analyses? If so, please indicate
Table 4: in table 4 appear the variables in the discriminant canonical models, the selected variables should be, as the authors described in the material and methods section, the significant variables of the stepwise discriminant analysis. But actually in the model with technical variables appeared sows and LU value, and sows was non-significant in the previous analyses, on the contrary did not appeared Dehesa land use, which was significant in the previous analyses. Authors should explain how the variables have been selected.
L295: delete “whole”
Author Response
REVIEWER 1
General comments
The paper deals with the study of differences in Iberian farms according to the fattening types based on technical-economic and environmental approach, as well as with the identification of the characteristics that can be reference points to differentiate the fattening types, and proposing strategies to improve sustainability of Iberian traditional pig production. The aim of the paper is interesting since the integration of technical-economic and environmental issues is necessary in the study of sustainability of livestock systems.
Taking into account the analyses performed and results of the paper I suggest to change the title of the paper. Because although some fattening strategies to improve sustainability of the systems were exposed by the authors at the end of the paper, in my opinion it was not the main objective of the paper.
The paper overall is sound and well written. The introduction section properly defines the problem and the proposed hypothesis taking into account related bibliography.
Methods have to be explained more in detail, it was not clear what was the sequence of the different analyses performed. Authors should clarify this section. Discussion in general is well written and succeed in discussing scientifically the observed findings in a biologically integrated fashion, both within the study as well as relative to results of other scientists.
AU: Dear anonymous Reviewer 1,
Thanks very much for your comments and for reviewing our manuscript in such detail! We have taken great care to acknowledge all your comments and to consider your suggestions for improvements to the manuscript. Please find below our responses to each of the points raised. The title has been changed accordingly. The new title is “Analysis of the sustainability of fattening systems for Iberian traditional pig production through a technical and environmental approach”.
Specific comments
Abstract
L35: “to study the relationships” instead of “to study of the relationships”
AU: Amended.
Material and Methods
L112: I suggest to move this sentence “Data were analysed to determine outliers to be discarded before analysis” to “statistical analyses section” and explain what statistic method was used to identify outliers.
AU: According to the referee the sentence has been moved to “statistical analyses section” and the test used is now included (new line 124).
L114: please describe all abbreviations the first time they appear in the text. Here and elsewhere
AU: Done throughout the manuscript.
Table 2: define abbreviations in the table footnote. In table 2 mean and SE was shown for the 36 studied farms. It would be of interest to show the mean and SE for each fattening type (MF and DF) and the number of farms in each group.
AU: Abbreviations are now included in Table 2. Regarding SE for each fattening type, we suppose that these suggestions are referred to Table 3. Therefore, they have been included in Table 3.
L139: please indicate the statistical programs or packages used for the discriminant analyses.
AU: The statistical programme and packages used is now indicated in the line 182. In this way, the information on all analyses is grouped together.
L139-140: “These techniques have been applied in previous studies with a focus on dairy sheep systems [25–27], meat goat systems [28] and fishery systems [29)” : this sentence is not adequate for the mat methods section .
AU: Amended (new line 140).
L141-146: I think this paragraph is not adequate for the mat methods section, maybe should be include to some extent in the discussion of the results.
AU: We consider this paragraph important for the understanding of the methodology by the reader with no knowledge of multivariate analysis. We have provided more information in lines 143-144.
L154: it is not clear the order of the performed analyses. In Line 138 the following order was assessed:” canonical discriminant analysis, stepwise discriminant analysis, and discriminant analysis.”, but here seems that first was performed the stepwise discriminant analysis. Please indicate the correct order for the three analyses.
AU: According to the reviewer, the sentence is clarified as follows: “The most discriminant variables obtained in the stepwise discriminant analysis were selected and used for the discriminant analysis” (new lines 163-164).
L157: how was it tested?
AU: The sentence is changed to clarify (new lines 164-165).
L165-174: In my opinion it is not necessary to explain what is a CCA analysis in the material and methods section.
AU: We consider this paragraph important for the understanding of the methodology by the reader with no knowledge of multivariate analysis. We would like to keep it as it is.
L174: Authors indicate the software but also should indicate the procedures.
AU: Procedures used are included in the lines 182-184.
There are much more technical variables than environmental ones, and the environmental variables are highly correlated between them (i.e. GW and GW value or LO and LO value). These differences could affect the capacity of the models to discriminate between fattening types?. Could exist and overestimation of the discriminant capacity of the environmental model?.
AU: Although all the variables have been analysed using CDA, the models have been developed using SDA in order to achieve the highest discriminant capacity with the fewest number of variables, and avoiding redundant information, which leads to overestimation. The best models in the three sets of variables contain only two variables (n=36) which, in the case of the environmental model, are not correlated (in the other two models, the correlation between both variables is less than 0.6). Although the discriminant capacity has not been validated with external cases, we believe that the existence of an overestimation is unlikely.
Results and discussion
177-180: please clarify if table 3 showed results of the canonical discriminant analysis or results of stepwise discriminant analysis.
AU: Table 3 shows the results of the canonical discriminant analysis. The variables with the greatest discriminating capacity are also indicated. The text has been modified accordingly in lines 188-189.
L184: Why is “sows” declared as a discriminant variable if the P-value in table 3 is not significant?
AU: There was an error in the table. The p-value is 0.014. So, the statement was correct.
Table 3 : define abbreviations in the table footnote. Every table and figure are independent and abbreviations should be described
AU: Amended.
L196-206: here authors described numerical changes, because these variables had not a significant P-value
AU: In lines 204-206 it is indicated that no differences are found in some variables, but these variables are very informative to characterise the different types of Iberian farms according to the results obtained.
L226-229: please do not discuss non-significant values
AU: Discussion has been removed.
L231: does it refers to the results of the canonical discriminant analyses? If so, please indicate
AU: The selection of the variables is based on the results of the different analyses in a combined approach.
Table 4: in table 4 appear the variables in the discriminant canonical models, the selected variables should be, as the authors described in the material and methods section, the significant variables of the stepwise discriminant analysis. But actually in the model with technical variables appeared sows and LU value, and sows was non-significant in the previous analyses, on the contrary did not appeared Dehesa land use, which was significant in the previous analyses. Authors should explain how the variables have been selected.
AU: The stepwise selection procedure is clarified in “material and methods” in agreement with the referee. Now we believe that this passage is much clearer in the lines 152-159.
L295: delete “whole”
AU: Amended (new line 302).

Reviewer 2 Report
Comments, review of manuscript id: Animals-1064757
Title: ”Fattening strategies to improve the sustainability of Iberian dehasa-based pig production through a technical and environmental approach”
The manuscript gives an analysis of the sustainability of two fattening systems for Iberian traditional pig production The manuscript needs some corrections, as mentioned in “Comments”.
The English language is in general good.
Comments.
In general: Be consequent when writing “Montanera farms” and “Diversified farms” in the text, tables and figures. In the tables and figures you sometime use the full manes, and sometime use the abbreviations “MF” and “DF”. Remember that tables and figures should be possible to read without reading the whole text.
Simple summary, line 22. Please specify the meaning of “this research” as ir is mentioned in this line.
Introduction, lines 72 – 74. Please specify your statement in this sentence. Your reference [18] represent a study made on Chinese conditions, how is the results relevant for the situation in Spain?
Materials and Methods, 2.2. Data acquisition, line 114. Please define your use of the abbreviations “CML” (or give a reference to “Center of Environmental Science of Leiden University”) and “LO” (“LO” is defined in “Table 2, but not in the text) as some readers may not be familiar to these abbreviations.
Materials and Methods, Table 2. When you refer to “kg Montanera” and “kg total pig production”, do you mean the live weight produced, or the kg of pork produced? Please make this clear for the reader.
Results and Discussion, Table 3. See my comments “In general”, as mentioned above. I suggest that you write both the full name and the abbreviations in the first lines of the table. Write “Montanera farms (MF)” and “Diversified farms (DF)”. Also, see my comments for Table 2, regarding “kg produced”.
Results and Discussion, Figure 2. See my comments “In general”, as mentioned above. I suggest that you write both the full name and the abbreviations in this figure, “Montanera farms (MF)” and “Diversified farms (DF)”.
Author Response
REVIEWER 2
Comments, review of manuscript id: Animals-1064757
Title: ”Fattening strategies to improve the sustainability of Iberian dehasa-based pig production through a technical and environmental approach”
The manuscript gives an analysis of the sustainability of two fattening systems for Iberian traditional pig production The manuscript needs some corrections, as mentioned in “Comments”.
The English language is in general good.
AU: Dear anonymous Reviewer 2,
Thank you for taking time and reading and for your valuable comments. We hope that this modified version will meet the expectations of the Reviewer.
Comments.
In general: Be consequent when writing “Montanera farms” and “Diversified farms” in the text, tables and figures. In the tables and figures you sometime use the full manes, and sometime use the abbreviations “MF” and “DF”. Remember that tables and figures should be possible to read without reading the whole text.
AU: Amended. Many Thanks for highlighting this point.
Simple summary, line 22. Please specify the meaning of “this research” as ir is mentioned in this line.
AU: Amended (new lines 22-23).
Introduction, lines 72 – 74. Please specify your statement in this sentence. Your reference [18] represent a study made on Chinese conditions, how is the results relevant for the situation in Spain?
AU: In the reference Wang et al. 2016, the authors studied the differences in the different pig systems due to a transformation from the traditional system to a more intensive system. The development of intensive management in the Iberian pig system is currently being implemented with considerable success. However, the reference has been changed to a European research (Asmild et al. 2006).
Materials and Methods, 2.2. Data acquisition, line 114. Please define your use of the abbreviations “CML” (or give a reference to “Center of Environmental Science of Leiden University”) and “LO” (“LO” is defined in “Table 2, but not in the text) as some readers may not be familiar to these abbreviations.
AU: The abbreviation CML has been removed to avoid confusion and the abbreviation LO is defined in line 114 in the text.
Materials and Methods, Table 2. When you refer to “kg Montanera” and “kg total pig production”, do you mean the live weight produced, or the kg of pork produced? Please make this clear for the reader.
AU: Amended.
Results and Discussion, Table 3. See my comments “In general”, as mentioned above. I suggest that you write both the full name and the abbreviations in the first lines of the table. Write “Montanera farms (MF)” and “Diversified farms (DF)”. Also, see my comments for Table 2, regarding “kg produced”.
AU: Amended.
Results and Discussion, Figure 2. See my comments “In general”, as mentioned above. I suggest that you write both the full name and the abbreviations in this figure, “Montanera farms (MF)” and “Diversified farms (DF)”.
AU: Amended.

Round 2
Reviewer 1 Report
The authors have adressed most of my previous comments. However I have only a few addditional comments.
In Table 3 results of the canonical discriminant analyses were presented. I have one doubt:why did appear with a "*" the variables selected from the stepwise discriminant analysis (footnote). It is confusing, if results from the canonical discriminant analyses are presented why results from the stepwise discriminant analysis are remarked in this table?. Or maybe table 3 presents the results of both analyses?. If so please clarify in L188-189.
About "3.2. Reference points in Iberian fattening production ": Is the canonical discriminant analyses repeated at this point only including the selected variables derived from the stepwise discriminant analyses?. If so, please clarify in material and methods and here. In L237 it should be noted "canonical discriminant analysis instead of discriminant analysis (as it was described in Table 4 and figure 1)
Author Response
The authors would like to thank the reviewers for their comments that have helped to improve the paper. We have marked in red colour the changes in the manuscript. Following a detailed answer to each comment.
We hope that this modified version will meet the expectations of the reviewer #1 and the editorial team.
The authors have adressed most of my previous comments. However I have only a few addditional comments.
AU: Dear anonymous Reviewer 1,
We have taken great care to acknowledge all your comments for improvements to the manuscript.
In Table 3 results of the canonical discriminant analyses were presented. I have one doubt: why did appear with a "*" the variables selected from the stepwise discriminant analysis (footnote). It is confusing, if results from the canonical discriminant analyses are presented why results from the stepwise discriminant analysis are remarked in this table?. Or maybe table 3 presents the results of both analyses?. If so please clarify in L188-189.
AU: It's a mistake. We corrected it in the text in the previous version, but we forgot to do it in the footnote. The asterisk indicates the most discriminating variables between types of fattening obtained by CDA. Now the footnote has been now modified.
About "3.2. Reference points in Iberian fattening production ": Is the canonical discriminant analyses repeated at this point only including the selected variables derived from the stepwise discriminant analyses?. If so, please clarify in material and methods and here. In L237 it should be noted "canonical discriminant analysis instead of discriminant analysis (as it was described in Table 4 and figure 1)
AU: The methodology (new lines 163-165) and the results section (new lines 239-240) have been modified according to the referee's indication.
